# Cannabinoid and Orexigenic Systems Interplay as a New Focus of Research in Alzheimer’s Disease

**DOI:** 10.3390/ijms25105378

**Published:** 2024-05-15

**Authors:** Joan Biel Rebassa, Toni Capó, Jaume Lillo, Iu Raïch, Irene Reyes-Resina, Gemma Navarro

**Affiliations:** 1Centro de Investigación Biomédica en Red Enfermedades Neurodegenerativas (CiberNed), National Institute of Health Carlos, 28029 Madrid, Spain; jrebaspa7@alumnes.ub.edu (J.B.R.); tonicapoquetglas@ub.edu (T.C.); lillojaume@gmail.com (J.L.); iuraichipanisello@gmail.com (I.R.); 2Institut de Neurociències UB, Campus Mundet, 08035 Barcelona, Spain; 3Department of Biochemistry and Physiology, Faculty of Pharmacy and Food Science, University of Barcelona, 08028 Barcelona, Spain; 4Departament de Biochemistry and Molecular Biomedicine, University of Barcelona, 08028 Barcelona, Spain

**Keywords:** Alzheimer, cannabinoid, orexigenic, heteromer

## Abstract

Alzheimer’s disease (AD) remains a significant health challenge, with an increasing prevalence globally. Recent research has aimed to deepen the understanding of the disease pathophysiology and to find potential therapeutic interventions. In this regard, G protein-coupled receptors (GPCRs) have emerged as novel potential therapeutic targets to palliate the progression of neurodegenerative diseases such as AD. Orexin and cannabinoid receptors are GPCRs capable of forming heteromeric complexes with a relevant role in the development of this disease. On the one hand, the hyperactivation of the orexins system has been associated with sleep–wake cycle disruption and Aβ peptide accumulation. On the other hand, cannabinoid receptor overexpression takes place in a neuroinflammatory environment, favoring neuroprotective effects. Considering the high number of interactions between cannabinoid and orexin systems that have been described, regulation of this interplay emerges as a new focus of research. In fact, in microglial primary cultures of APPSw/Ind mice model of AD there is an important increase in CB_2_R–OX_1_R complex expression, while OX_1_R antagonism potentiates the neuroprotective effects of CB_2_R. Specifically, pretreatment with the OX_1_R antagonist has been shown to strongly potentiate CB_2_R signaling in the cAMP pathway. Furthermore, the blockade of OX_1_R can also abolish the detrimental effects of OX_1_R overactivation in AD. In this sense, CB_2_R–OX_1_R becomes a new potential therapeutic target to combat AD.

## 1. Introduction

Alzheimer’s disease (AD) was initially documented by Alois Alzheimer in 1906 and currently constitutes 60% to 80% of the 55 million cases of dementia worldwide [1]. AD manifests in two forms: sporadic and familial. The sporadic form is characterized by an incompletely understood etiopathogenesis, including lifestyle, environment, and genetic and epigenetic alterations, and accounts for approximately 95% of AD cases. Conversely, familial AD arises from mutations in genes such as amyloid precursor protein (APP), presenilin-1, and presenilin-2, and constitutes around 5% of AD cases [2].

Alzheimer’s disease begins with the impairment of new memory retention and advances through diverse cognitive and behavioral alterations, intensifying over time. The pathophysiological mechanism of AD is typified by aberrations in amyloid precursor protein (APP) cleavage, instigating the production of the APP fragment beta-amyloid (Aβ). Concurrently, there is hyperphosphorylation of tau protein, culminating in the formation of aggregates that elicit vascular, metabolic, and inflammatory perturbations [3], precipitating synaptic dysfunction, neuronal demise, and typically culminating in gradual neurodegeneration.

Despite extensive research efforts, AD currently lacks a definitive cure, and the available treatments only address the symptoms of the disease. These treatments include acetylcholinesterase inhibitors [4] and the NMDAR inhibitor memantine [5]. The absence of a cure for AD, coupled with the enigmatic nature of its etiology, underscores the urgent need for the identification and pursuit of novel therapeutic targets. Along this line, orexigenic (OXR) and cannabinoid (CBR) receptors are of interest because of their contribution to promising aspects of drug discovery demonstrated by the plethora of articles published in the last few years [6,7,8]. Evidence for functional interactions and/or heterodimerization between orexin receptors and cannabinoid receptors is discussed in the context of its relevance to AD.

## 2. GPCRs and Alzheimer’s Disease

G protein-coupled receptors (GPCRs) constitute the largest family of cell-surface receptors, and are encoded by more than 800 genes in the human genome [9]. These are seven-transmembrane-domain proteins that regulate a diverse array of intracellular signaling cascades in response to hormones, neurotransmitters, ions, photons, odorants and other stimuli [10]. The orthosteric ligand-binding site of a GPCR is located either among transmembrane helices or within the extracellular domain, while the transmembrane 6th, the 3rd intracellular loop and the carboxy-terminal end are involved in G protein coupling [11].

As such, GPCRs play an essential role in physiology and disease by triggering intracellular signaling through coupling to G proteins and ß-arrestins. In this sense, nowadays, GPCRs are the major targets of drugs in clinical use [12]. Nevertheless, since the 1990s, numerous reports have successively shown oligomerization of GPCRs and it is now widely accepted that oligomerization is a universal aspect of GPCR biology [13]. The formation of GPCR complexes is associated with a diversification in their coupling to different G proteins, as well as in their ligand binding regulation, ontogeny and desensitization properties. In this sense, oligomeric units have functional characteristics that differ from those of their constituent receptors in their monomeric form [14]. Over the years, the elucidation of GPCR targets from neural circuits has been shown to be a promising method with which to treat disorders pertaining to the nervous system and extending their scope to other systems has proved to be of interest to researchers. In fact, several studies have presented compelling evidence implicating GPCRs in multiple stages of the pathogenesis of Alzheimer’s disease (AD). 

Research using magnetic resonance imaging (MRI) has revealed that a decreased volume in the hippocampus and entorhinal cortex, regions affected early in the progression of the disease [15], as well as a reduced cortical thickness in areas such as the medial temporal, inferior temporal, temporal pole, angular gyrus, superior parietal, superior frontal, and inferior frontal cortex, correspond with the cognitive impairments observed in patients with AD [16]. Furthermore, changes in several neurochemical pathways, including the acetylcholine, serotonin, adenosine, orexin (OX) and cannabinoid signaling pathways have been shown to be involved in memory loss observed in AD [17,18,19]. 

The pathophysiology of AD involves the aggregation of extracellular amyloid-β, inducing neuronal degeneration, and the aberrant phosphorylation of tau protein, promoting the assembly of neurofibrillary tangles. These tangles disrupt the typical functionality of neurons and synapses, causing neuronal death and neuroinflammation [20]. Amyloid-β is produced from sequential proteolysis of the APP by β- and γ-secretase. By contrast, sequential cleavage by α- and γ-secretase precludes amyloid-β generation. α, β- and γ-secretases are regulated by GPCRs, and amyloid-β itself can directly or indirectly affect GPCR function [21,22]. Indeed, there is evidence that GPCRs can bind to β-secretase (β-site APP cleaving enzyme 1, BACE1) and γ-secretase [23]. There is currently no cure for AD, and new potential treatments do not usually succeed, as happened with inhibitors targeting BACE1 and γ-secretase. These inhibitors have not been approved by the FDA due to the fact that they are not specific enough as they can also inhibit the normal biological functions of secretases [24]. Therefore, the exploration of novel therapeutic targets focused on GPCRs, including biased ligands and allosteric modulators, present a promising avenue for preserving efficacy and managing adverse effects [25]. The orexin/hypocretin system has been implicated in many pathways, and its dysregulation is under investigation in several diseases. Disorders in which orexinergic mechanisms are being investigated include narcolepsy, idiopathic sleep disorders and neurodegenerative diseases such as AD [26]. 

## 3. Involvement of Orexin System in Alzheimer’s Disease

Orphan GPCRs, which are GPCRS whose endogenous ligands have not yet been elucidated, are regarded as promising targets for drug development [27]. Therefore, the task of de-orphanizing (identifying the endogenous ligand of) these orphan GPCRs has been actively pursued. In 1998, the orexin receptor was de-orphanized [28]. During their search for potent ligands in rat brain extracts that target orphan GPCRs, Sakurai et al. discovered two hypothalamic peptides exhibiting specific affinity for a particular orphan GPCR family. These peptides were named orexin A (OXA) and orexin B (OXB) because of their ability to stimulate appetite when administered intracerebroventricularly [29]. Orexin A, composed of 33 amino acids with 2 disulfide bonds, and orexin B, composed of 28 amino acids, correspond with hypocretin 1 and hypocretin 2, respectively [30]. 

The orexin system presents a high degree of conservation across mammalian species. The involvement of this system in narcolepsy has significantly promoted the study of orexin genetics [31]. Orexins are encoded by a common precursor polypeptide named prepro-orexin, which is synthesized by a cluster of neurons situated in the lateral hypothalamus (LH) and adjacent brain regions [32]. The functions of these peptides are enabled by two distinct receptors belonging to the class A rhodopsin-like subfamily of GPCRs. These receptors are known as orexin receptor type 1 (OX_1_R) and type 2 (OX_2_R) [6], which consist of 425 and 444 amino acids, respectively, and share 64% amino acid identity [33]. While orexin-B mainly targets OX_2_R, OXA binds to both receptors and shows an affinity for OX_1_R that is between 5 and 100 times higher than orexin B [34]. 

The binding of orexins to their receptors primarily triggers the activation of mainly Gq/11 protein, but Gs and Gi/o proteins have also been observed, as well as the recruitment of β-arrestins [6]. This activation initiates various downstream signaling pathways, engaging effectors such as protein kinases, ion channels, and phospholipases, such as phospholipase C (PLC), that elevate intracellular Ca^2+^ levels. It is noteworthy that calcium ions (Ca^2+^) play a vital role as a second messenger in OXR signaling [6]. Nevertheless, orexin receptors are considered promiscuous receptors because the sum of studies regarding their G protein preference is inconclusive [35]. 

Both OX_1_R and OX_2_R exhibit widespread distribution throughout the central nervous system (CNS), being located in the ventral tegmental area (VTA), hypothalamus, lateral hypothalamic area, amygdala, and midbrain. Additionally, OX_1_R shows a preference for the locus coeruleus (LC), while OX_2_R is primarily expressed in the tuberomammillary nucleus (TMN) [36]. Thus, the varying distribution patterns of OX_1_R and OX_2_R, along with the differing subtype-selectivity of the orexin peptides for these receptors, could potentially explain the unique physiological effects observed within the orexin/receptor pathways [6]. Considering the extensive expression of OXRs in CNS and peripheral regions and their involvement in numerous physiological functions, the orexigenic system has been shown to have significant implications in various pathological conditions such as sleep–wake dysregulation, panic and mood disorders, obesity, metabolic syndrome, ischemia, and chronic inflammation, multiple sclerosis and intestinal bowel disease, cancer, and also Alzheimer’s disease (AD) [6,7,8,9,10,11,12,13,14,15,16,17,18,19,20,21,22,23,24,25,26,27,28,29,30,31,32,33,34,35,36,37]. The dysregulation of the sleep–wake cycle is often linked to both cognitive and behavioral deficits in AD. In fact, a consistent sleep–wake cycle serves a beneficial role in safeguarding synaptic plasticity and brain function against the neuropathological damage caused by neurofibrillary degeneration and the accumulation of Aβ plaques [38]. Notably, biomarkers found in cerebrospinal fluid (CSF) such as Aβ42, Aβ40, total tau, and phosphorylated tau show significant associations with decreased nocturnal sleep patterns [39]. 

Moreover, recent research has proposed that amyloid deposition influences memory through its impact on sleep patterns; however, the precise mechanisms through which amyloid affects sleep and subsequently disrupts memory functions remain incompletely understood [40]. The orexigenic system represents a potentially significant mediator that could elucidate this ambiguity. Though there are contradicting results regarding the orexin levels in AD patients [41], there is a crucial link between orexins and AD. Indeed, the importance of orexins in the context of the pivotal role of the sleep–wake cycle in the progression of Alzheimer’s disease is frequently evidenced. Orexins play a crucial role in the promotion of wakefulness in mammals by modulating neuronal activity, which ceases during sleep [42]. There is a growing focus on the investigation of the mechanisms underlying the sleep–wake cycle and its influence on the levels of Aβ in the interstitial fluid of the brain. Soluble Aβ is a significant contributor to neurotoxicity, leading to synaptic impairment and dysfunction. The regulation of soluble Aβ clearance is thought to be significantly influenced by the dynamics of the sleep–wake cycle [43]. The orexinergic signaling pathway is involved in regulating hippocampal circadian oscillations, potentially exerting a substantial influence on the expression of genes that play crucial roles in the production and transport of Aβ [44]. Excessive orexinergic signaling is suggested to contribute to disruptions in the sleep–wake cycle, thereby accelerating Aβ deposition [45]. In contrast, a study has demonstrated that orexin impairs the degradation of Aβ by microglia, further implicating orexin in the pathological mechanisms underlying AD [46].

## 4. Role of Endocannabinoid System in Alzheimer’s Disease

The endocannabinoid system (ECS) is the most widespread GPCR system in the brain. Endocannabinoids are engaged in the regulation of many physiological functions, including analgesia, appetite, learning and memory [47]. Cannabis was one of the first plants used as a drug, first in religious ceremonies and finally recreationally. The first accounts of its use for these purposes stretch back 5000 years [48]. CBD was first isolated from marijuana in 1940 [49], however, CBD was neglected and eclipsed by (−)-Δ^9^-tetrahydrocannabinol (Δ^9^-THC), the main psychoactive phytocannabinoid, isolated in 1964 [50]. 

The endocannabinoid system is constituted by cannabinoid receptors type 1 and 2 (CB_1_R and CB_2_R), their endogenous ligands, and the enzymes involved in synthesizing and degrading endocannabinoids. The different elements of this system are very abundant in the brain, where they modulate multiple functions, such as synaptic transmission, immunological responses, and cell proliferation [51]. CB_1_ and CB_2_ receptors, which belong to the family of seven transmembrane G protein-coupled receptors, are able to bind endocannabinoid ligands, being the derivatives of the fatty arachidonic acid, anandamide (AEA) and 2-arachidonoylglycerol (2-AG), the most abundant and well described [49]. It is necessary to highlight that the ECS is considered to be a multifunctional system. Despite being an isolated system, it influences, and is influenced by, many other signaling pathways. In this sense, the ECS has recently attracted significant interest as a potential target for novel drug development related to neurodegenerative diseases [52].

The CB_1_ cannabinoid receptor was discovered and subsequently cloned on the basis of its responsiveness to Δ^9^-THC [53]. The CB_1_R was shown to be activated by the endocannabinoid anandamide [54], and this was followed by the identification of a second metabolite, the 2-arachidonoylglycerol (2-AG) [55]. The CB_2_ cannabinoid receptor was then identified in 1993. CB_2_R expression in the CNS is less established when compared with CB_1_R, as the presence of CB_2_R on neurons has been a subject of debate. However, evidence demonstrates its expression at low levels in dopaminergic neurons, glial cells, and endothelial cells [56,57]. CB_1_R and CB_2_R exhibit seven transmembrane domains, an extracellular N-terminus, and an intracellular C-terminus, showing as 44% homologous overall and as 68% homologous within the transmembrane domains [58]. 

Endocannabinoid system alterations have been shown to be involved in the pathophysiology of AD [59]. On the one hand, in mouse models of AD, the capacity of the CB_1_R agonist to inhibit the excessive release of glutamate, and thereby reduce excitotoxicity and neuronal damage, has been demonstrated [60]. On the other hand, CB_2_R has shown neuroprotective effects by promoting amyloid clearance and moderating microglial recruitment [61]. CB_2_ receptors have been identified in the postmortem brain tissue of patients with Alzheimer’s disease, where these receptors were substantially and selectively expressed in neuritic plaque-associated microglia [62]. CB_2_ receptors are also upregulated in reactive microglial cells in animal models of Alzheimer’s disease [63,64]. Overall, CBRs have been shown to be potential therapeutic targets in models of diseases with limited or no currently approved therapies, such as Alzheimer’s disease [51,64].

## 5. Interactions between Orexigenic and Endocannabinoid Systems

In the last two decades, endocannabinoid and orexin interdependence has become a topic of growing interest not only for their individual roles, but also for the physical interaction that has been described between receptors of both systems [65]. Cannabinoid and orexin receptors are both found in many of the brain regions underlying complex behaviors such as sleep, appetite, and reward processing. Cellular and molecular interactions between these neuromodulatory systems have physiological implications in homeostasis, neurological and psychiatric disorders as well as in drug–drug interactions between cannabinoid and orexin compounds [66]. 

It has been observed that 2-AG synthesis is stimulated in response to OX_1_R stimulation [67] (Figure 1). Endocannabinoids are lipid-based signaling molecules exhibiting distinctive pharmacological profiles and regional distribution [68]. Unlike typical neurotransmitters, they are not stored in presynaptic vesicles but are synthesized and released upon demand in postsynaptic terminals in response to increases in intracellular Ca^2+^ levels [69]. These compounds serve as retrograde synaptic messengers traversing synapses and binding to presynaptic and postsynaptic cannabinoid receptors [69] and to receptors located in glial cells [70], thereby modulating neurotransmitter release. CB_1_R is detected in the hypothalamus, localized presynaptically on GABAergic neurons and on glutamatergic projections to OX neurons [71]. Thus, 2-AG mediates the effects of OXs on nociception, addiction and food intake [35,67]. Furthermore, it has been shown that the administration of the CB_1_R inverse agonist, rimonabant, blocks the orexigenic effect of OXA administered by intracerebroventricular route [72]. Additionally, intracerebroventricular injection of the CB_1_R inverse agonist produced a significant decrease in the number of neurons expressing OXA in the hypothalamus [73]. Thus, it seems that the ECS may influence food intake by regulating the expression and/or action of OXs. On the other hand, the elevation of dopamine extracellular levels in the nucleus accumbens induced by Δ^9^-THC is inhibited in mice lacking OX_1_R, indicating that cannabinoids rely on orexinergic transmission to modulate the dopaminergic mesolimbic pathway [74]. For this reason, acutely activating both cannabinoid and orexin receptors increases appetite [75] and reward sensitivity [76] (Figure 1).

However, regarding sleep and arousal, as cannabinoid and orexin systems are modulated by other neuronal populations different from those regulating appetite and reward, the regulation here is different. It has been observed that cannabinoids reduce the activity of OX receptors in orexinergic neurons due to the presynaptic attenuation of neurotransmitter release, leading to a decrease in arousal [26,71]. In fact, orexin receptor antagonists are emerging as effective treatments for insomnia [77], while phytocannabinoids are used as sleep aids that affect the sleep–wake neuropathway [77]. Thus, cannabinoid and orexin receptors may potentiate one another in brain regions involved in appetite and reward, while having antagonistic functions in regions underlying arousal and sleep [66] (Figure 1).

Beyond the physiological interaction between both systems, new evidence is being obtained to demonstrate physical interactions between cannabinoid and orexin receptors [78]. OX_1_R and OX_2_R have the capability to form constitutive homo and heterodimers not only with each other but also with various GPCRs, including cannabinoid CB_1_ and CB_2_ receptors [78,79] (Figure 1). The co-localization of CB_1_R with both OX_1_R and OX_2_R has been found in many brain regions such as the neocortex, hippocampus, thalamus, hypothalamus, amygdala and the ventral tegmental area [66]. Since the 1990’s it has been well known that the formation of GPCR heteromeric complexes appears to activate distinct signaling pathways from those initiated by the constituent monomers, resulting in diverse physiological effects [80]. The intercommunication between protomers is a consequence of a defined quaternary structure, responsible for the specific functional characteristics of the heteromer [81]. The modulation between receptors of a GPCR heteromer can be observed as crosstalk or as cross-antagonism phenomena, among others, changing the functionality of the heteromer in comparison with the receptor activity [82]. Agonist binding induces cross-conformational changes between receptor protomers and GPCR-associated proteins, including heterotrimeric G proteins and β-arrestins [82]. In this regard, new structural complexes constituted by receptors sensing the same ligand but producing opposite signaling effects, can emerge as novel interesting pharmacological targets [82].

For this reason, new studies have focused on the physical interactions between cannabinoid and orexin receptors and on the functional consequences of these heteromerizations. Regarding the CB_1_R–OX_1_R heteromeric complex, evidence suggests a crosstalk between the cannabinoid receptor CB_1_ and OX_1_R in a heterologous system. Moreover, when the two receptors are forming complexes, a major OXA potency to activate the MAPK pathway through CB_1_R-dependent enhancement is observed, as dose–response curves have indicated a 100-fold increase in the potency of orexin-mediated MAPK activation compared with cells expressing only OX_1_R [83] (Figure 1). This effect requires a functional CB_1_ receptor, as evidenced by the blockade of the orexin response by the specific CB_1_ receptor inverse agonist rimonabant (SR141716), but also by pertussis toxin, suggesting that this potentiation is Gi-mediated [84]. In contrast with OX_1_R, the potency of the direct activation of CB_1_R is not affected by co-expression with OX_1_R. These data provide evidence that CB_1_ receptor is able to potentiate OX_1_R signaling. Considering the anti-obesity effect of SR141716, these results can help one to understand the mechanism by which the molecule may prevent weight gain through functional interaction between CB_1_R and other receptors involved in the control of appetite [83]. In parallel, it has been demonstrated that the internalization of the cannabinoid CB_1_ receptor can be induced by the peptide orexin A when the orexin OX_1_ receptor is co-expressed along with the cannabinoid CB_1_ receptor. Remarkably, OXA was substantially more potent in producing internalization of the CB_1_ receptor than in causing internalization of the bulk OX1 receptor population [85,86]. This process could function as a negative feedback mechanism that avoids an excessive CB_1_R overactivation and thus, an exacerbated orexin response. 

However, when studying the CB2R–OX1R heteromeric complex, it has been demonstrated that the OX_1_R antagonist SB334867 not only blocks the OXA-induced effect, but also potentiates cannabinoid CB_2_R function in the cAMP and MAPK signaling pathways [79]. Considering that each receptor complex shows a distinct heteromeric print, more studies to determine the functionality of all of the different complexes formed between cannabinoid and orexin receptors should be performed. Taking into account the differences mentioned between OX_2_R and OX_1_R regarding their expression and functionality in the CNS, it would be interesting to study the effects of OX_2_R antagonism in the OX_2_R–CB_1_R and OX_2_R–CB_2_R heteromers as well.

## 6. Orexin–Cannabinoid Heteromer Formation as a Potential Target to Combat Alzheimer’s Disease

As mentioned above, significant progress has been made in understanding the importance of the quaternary structure of GPCRs. Despite being believed to exist and to function as monomers for many years, it is now firmly established that most GPCRs exist as dimers and/or in higher order oligomers [87,88,89,90], and that this quaternary structure is necessary for plasma membrane expression and function [91]. 

The potential role of GPCR heteromers in Alzheimer’s disease has been described [92,93,94]. For example, there is evidence that, in primary cultures of microglia, blockade of A_2A_ receptors results in increased CB_2_R-mediated signaling. This heteromer-specific feature suggests that A_2A_R antagonists would potentiate, via microglia, the neuroprotective action of endocannabinoids, with implications for AD therapy [95]. In fact, the neuroprotective effect of CB_2_R regulating microglial activation and polarization in Alzheimer’s disease is highly described in the literature [96,97,98]. The mechanisms underlying the promotion of microglial polarization toward a neuroprotective phenotype, commonly referred to as M2, remain poorly understood, although certain GPCRs have been implicated in their regulation. Specifically, cannabinoids acting on the CB_2_ receptor have been suggested to confer neuroprotection and impede the advancement of neurodegenerative conditions [99,100]. However, as previously mentioned, the presence of cannabinoid receptors when forming oligomers with OXR can enhance orexin receptor action [83], which could have a negative impact on the prognosis of Alzheimer’s disease. Considering that the blockade of OX_1_R leads to CB_2_R signaling potentiation, the use of orexin receptor antagonists becomes a therapeutic strategy of interest for preventing the harmful effects mediated by OXR, while promoting the beneficial effects of CBR [79]. 

As mentioned earlier, both direct and indirect interactions between the orexigenic and cannabinoid systems can occur in various structures of the CNS (Figure 1). Specifically, it has been observed that physical interactions between receptors which lead to the formation of OXR–CBR heteromers predominate in brain regions most affected in AD, such as the hippocampus [66]. In this line, the findings elucidated by Raïch and colleagues indicate that the CB2R–OX1R complex is overexpressed in microglia from AD animal models [79]. Interestingly, the OX_1_R antagonist, SB334867, not only inhibits the orexin A-induced response but also potentiates cannabinoid CB_2_R function in cAMP and MAPK signaling cascades. Furthermore, this enhancement is more pronounced in activated microglial cells when compared with quiescent cells. Hence, OX_1_R antagonists present a promising novel therapeutic target for AD and other neurodegenerative disorders characterized by microglial activation [79]. 

In this manner, we hypothesize that the use of OX_1_R antagonists within the heteromer that is formed by OX_1_R and CB_2_R could improve AD prognosis through various processes. Excessive orexinergic signaling is suggested to contribute to disruptions in the sleep–wake cycle, thereby accelerating Aβ deposition. Consequently, the blockade of OX_1_R hyperactivation could potentially restore the sleep–wake cycle [101] and consequently reduce Aβ accumulation [102]. Blocking OX_1_R enhances the neuroprotective effect of CB_2_R, leading to increased activation of the neuroprotective phenotype of microglia, thereby preventing the neuroinflammatory process [79]. Thus, the potentiation of CB_2_R and the reduction of the proinflammatory phenotype would prevent the secretion of proinflammatory molecules, thereby halting the excitotoxic process (Figure 2).

Finally, the implication of OX_1_R, specifically in the rewarding properties of cannabinoids, though not in other pharmacological effects, holds particular interest, as OX_1_R antagonists might prove beneficial in treating cannabis dependence without interfering with other advantageous properties of cannabinoids in AD [103]. The development of innovative tools for deeper investigation into this functional interaction may unveil novel therapeutic strategies for neurological disorders involving these receptors. 

## 7. Conclusions

The convergence of orexins and cannabinoids in the context of AD is an evolving field. Orexin antagonists targeting the OXR–CBR heteromers are emerging as new compounds due to their potential to revert AD symptomatology. Continued research into their molecular and behavioral interactions promises to provide deeper insights into their significance in both physiological and pathological states, thereby offering potential therapeutic avenues. For this reason, the study of the heteromers formed by OXR–CBR becomes a new focus of research in finding new potential therapeutic targets for this disease.

To demonstrate the physiological role of the heteromer formed between cannabinoid and orexin receptors and validate the idea that orexin receptors regulate cannabinoid receptors within these complexes, it would be interesting to see how this regulation is lost in an orexin receptor KO animal. We propose in vitro testing of inflammation markers, oxidative stress and AB presence/transport using primary cultures of neurons and microglia of the orexin receptor KO animals. Then, in vivo experiments to test cognitive impairments and sleep–wake cycles could be performed.

## Figures and Tables

**Figure 1 ijms-25-05378-f001:**
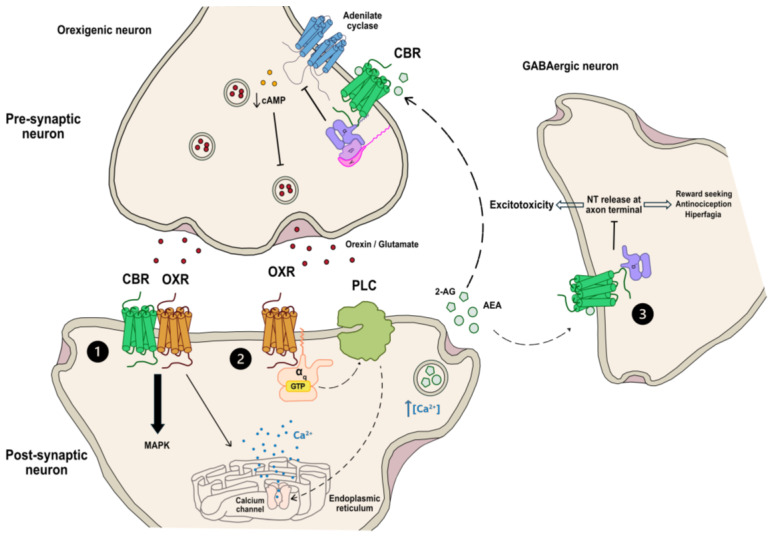
Interplay between the cannabinoid and orexigenic systems. This diagram illustrates the different pathways through which cannabinoid and orexigenic systems interact with each other. The interactions can be as follows: direct through the formation of heteromers (**1**), or indirect via other components of both systems (**2** and **3**). When forming heteromers, both receptors (CB_1_R–OX_1_R) mutually potentiate each other, resulting in an intracellular increase in calcium through a Gq-protein-phospholipase C cascade mediated by OXR and an enhancement of the MAPK pathway mediated by both receptors (**1**). Upon orexin release, activation of postsynaptic OXR triggers the synthesis of the endocannabinoid 2 arachidonoylglycerol (2-AG) through a signaling cascade involving Gq-protein and phospholipase C-diacylglycerol lipase (PLC-DAGL). Subsequently, 2-AG diffuses into the extracellular space and binds to CBR on presynaptic cells (**2**) or neighboring cells (**3**), inducing hyperpolarization of neuronal membranes and suppressing neurotransmitter release at axon terminals. When neighboring cells are GABAergic interneuron cells, this can avoid its inhibitory role and promote a higher neuronal activation (**3**) inducing an excitotoxity state.

**Figure 2 ijms-25-05378-f002:**
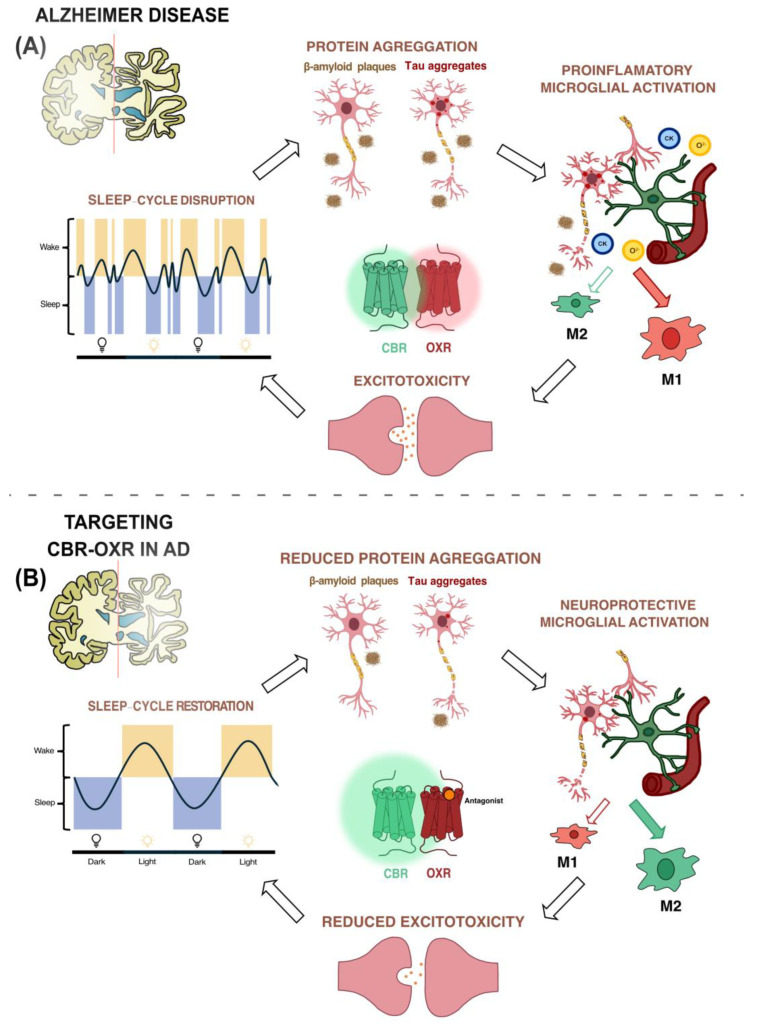
CBR–OXR heteromeric complex as a potential therapeutic target by which to reverse the classical symptoms in Alzheimer’s disease. (**A**) In AD, activation of orexin receptors promotes dysregulation of the sleep–wake cycle, which is associated with increased accumulation of Aβ and tau proteins, accelerating disease progression. Consequently, this leads to neuronal death and microglial activation, releasing proinflammatory molecules and triggering glutamate release to counteract this adverse situation. Overall, this neuroinflammatory state promotes dysfunction of orexin receptors, creating a cycle that worsens the disease prognosis. (**B**) CB_2_R–OX_1_R heteromers hold great potential for treating AD symptoms. Treatment with OX_1_R antagonists enhances microglial neuroprotective functions, mitigating the neuroinflammatory process through the OX_1_R–CB_2_R heteromer. Reduced production of proinflammatory molecules by microglia and the consequent decrease in excitotoxicity ensure a state of neuroprotection. Furthermore, OX_1_R blockade in the OX_1_R–CB_2_R heteromer potentiates the neuroprotective effects of CB_2_R, further enhancing the beneficial effects of this drug.

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
