# Peer review of "Cannabinoid and Orexigenic Systems Interplay as a New Focus of Research in Alzheimer’s Disease"

_ijms, 2024, doi:10.3390/ijms25105378_

Round 1

Reviewer 1 Report

Comments and Suggestions for Authors

Brilliant document.

As with all excellent reviews it raises a lot of questions. I had difficulty coming to terms with what experiments would push along this emerging field of research. What do we need to do to push ahead?

As a suggestion to the authors, can they provide a brief section with a flow chart on type(s) of experiments you would design using OR1- or OR2 receptor knockout mice to test the role of these receptors in Alzheimer's disease and the modulatory effects of cannabinoids?

In addition to or alternatively, can the authors briefly describe experiments that in vivo researchers can do in the near future to further this great concept? This would include studies in humans, too??

Minor

Spell out GPCR in the Abstract

Author Response

Brilliant document.

As with all excellent reviews it raises a lot of questions. I had difficulty coming to terms with what experiments would push along this emerging field of research. What do we need to do to push ahead?

As a suggestion to the authors, can they provide a brief section with a flow chart on type(s) of experiments you would design using OR1- or OR2 receptor knockout mice to test the role of these receptors in Alzheimer's disease and the modulatory effects of cannabinoids?

In addition to or alternatively, can the authors briefly describe experiments that in vivo researchers can do in the near future to further this great concept? This would include studies in humans, too??

Answer: thanks for the advice, we have included a paragraph with this information.

Minor

Spell out GPCR in the Abstract

Answer: thank you for your comment, this has been addressed

Reviewer 2 Report

Comments and Suggestions for Authors

The manuscript by Rebassa et al titled "Cannabinoid and Orexigenic Systems Interplay as a New Focus of Research in Alzheimer's Disease" addresses the interaction between the cannabinoid and orexigenic systems as a new focus of research in Alzheimer's disease. The manuscript explores how the interaction between cannabinoid and orexigenic receptors, specifically the CB2R-OX1R heteromeric complex, may represent a promising therapeutic target to combat the disease.

Aspects that need to be improved in the manuscript:

- In the abstract, the focus on the interaction between the cannabinoid and orexigenic systems in Alzheimer's disease is concisely introduced, highlighting the potential of the therapeutic target CB2R-OX1R, but it would be interesting for the authors to mention the main results that support the thesis of the article to increase the informative value of the summary for readers who would like to read the article in full.

- In the introduction, the justification for focusing on GPCRs, particularly cannabinoid and orexigenic receptors, needs to be expanded to directly link the introduction to the research focus. More recent articles could be cited.

- Results on the interaction between the cannabinoid and orexigenic systems are described in a narrative way, so it would be necessary to include tables of summarized data or they could be figures that directly support the statements made in the manuscript, such as changes in the expression of receptors or functional assays demonstrating therapeutic effects .

- In the discussion, the limitations of the study and how these may impact the interpretation of the results need to be addressed. Furthermore, alternative explanations for the observed effects could be discussed to strengthen the objectivity of the discussion.

Comments on the Quality of English Language

The quality of English is adequate

Author Response

The manuscript by Rebassa et al titled "Cannabinoid and Orexigenic Systems Interplay as a New Focus of Research in Alzheimer's Disease" addresses the interaction between the cannabinoid and orexigenic systems as a new focus of research in Alzheimer's disease. The manuscript explores how the interaction between cannabinoid and orexigenic receptors, specifically the CB2R-OX1R heteromeric complex, may represent a promising therapeutic target to combat the disease.

Aspects that need to be improved in the manuscript:

- In the abstract, the focus on the interaction between the cannabinoid and orexigenic systems in Alzheimer's disease is concisely introduced, highlighting the potential of the therapeutic target CB2R-OX1R, but it would be interesting for the authors to mention the main results that support the thesis of the article to increase the informative value of the summary for readers who would like to read the article in full.

Answer: thank you, the abstract has been modified.

- In the introduction, the justification for focusing on GPCRs, particularly cannabinoid and orexigenic receptors, needs to be expanded to directly link the introduction to the research focus. More recent articles could be cited.

Answer: thank you, more references have been added.

- Results on the interaction between the cannabinoid and orexigenic systems are described in a narrative way, so it would be necessary to include tables of summarized data or they could be figures that directly support the statements made in the manuscript, such as changes in the expression of receptors or functional assays demonstrating therapeutic effects.

Answer: thank you for the comment, a summary of the results explained in the text can be found in figure 2.

- In the discussion, the limitations of the study and how these may impact the interpretation of the results need to be addressed. Furthermore, alternative explanations for the observed effects could be discussed to strengthen the objectivity of the discussion.

Answer: Thank you for your comment, we have included in the conclusions section a paragraph suggesting future experiments to solve the limitations of the study.

Round 2

Reviewer 2 Report

Comments and Suggestions for Authors

All suggestions indicated for the authors have been implemented, thus I consider that the article could be accepted for publication

Comments on the Quality of English Language

The writing is adequate with minor corrections